# Changes in the Expression of TGF-Beta Regulatory Pathway Genes Induced by Vitamin D in Patients with Relapsing-Remitting Multiple Sclerosis

**DOI:** 10.3390/ijms241914447

**Published:** 2023-09-22

**Authors:** Alberto Lozano-Ros, María L. Martínez-Ginés, José M. García-Domínguez, Sara Salvador-Martín, Haydee Goicochea-Briceño, Juan P. Cuello, Ariana Meldaña-Rivera, Yolanda Higueras-Hernández, María Sanjurjo-Sáez, Luis A. Álvarez-Sala-Walther, Luis A. López-Fernández

**Affiliations:** 1Servicio de Neurología, Instituto de Investigación Sanitaria Gregorio Marañón, Hospital General Universitario Gregorio Marañón, 28007 Madrid, Spain; mluisa.martinez@salud.madrid.org (M.L.M.-G.); jmanuel.garcia@sen.es (J.M.G.-D.); haydee.goicoechea@salud.madrid.org (H.G.-B.); juanpablo.cuello@salud.madrid.org (J.P.C.); arimelda@ucm.es (A.M.-R.); yyhiguera@ucm.es (Y.H.-H.); 2Servicio de Farmacia, Instituto de Investigación Sanitaria Gregorio Marañón, Hospital General Universitario Gregorio Marañón, 28007 Madrid, Spain; sara.salvador@iisgm.com (S.S.-M.); maria.sanjurjo@salud.madrid.org (M.S.-S.); 3Servicio de Medicina Interna, Instituto de Investigación Sanitaria Gregorio Marañón, Hospital General Universitario Gregorio Marañón, 28007 Madrid, Spain; laalvare@ucm.es

**Keywords:** vitamin D, TGF-β pathway, BMP signaling, multiple sclerosis

## Abstract

Vitamin D is an environmental factor related to multiple sclerosis that plays a significant role in immune regulation. TGF-β is a superfamily of cytokines with an important dual effect on the immune system. TGF-β inhibits the Th1 response while facilitating the preservation of regulatory T cells (FOXP3+) in an immunoregulatory capacity. However, when IL-6 is present, it stimulates the Th17 response. Our aim was to analyze the regulatory effect of vitamin D on the in vivo TGF-β signaling pathway in patients with relapsing-remitting multiple sclerosis (RRMS). A total of 21 patients with vitamin D levels < 30 ng/mL were recruited and supplemented with oral vitamin D. All patients were receiving disease-modifying therapy, with the majority being on natalizumab. Expression of *SMAD7*, *ERK1*, *ZMIZ1*, *BMP2*, *BMPRII*, *BMP4*, and *BMP5* was measured in CD4+ lymphocytes isolated from peripheral blood at baseline and one and six months after supplementation. *SMAD7* was overexpressed at six months with respect to baseline and month one. *ERK1* was overexpressed at six months with respect to month one of treatment. No significant differences in expression were observed for the remaining genes. No direct correlation was found with serum vitamin D levels. *BMPRII* expression changed differentially in non–natalizumab- versus natalizumab-treated patients. Changes were observed in the expression of *ERK1*, *BMP2*, and *BMP5* based on disease activity measured using the Rio-Score, *BMP2* in patients who had relapses, and *BMP5* in those whose EDSS worsened. Our results suggest indirect regulation of vitamin D in TGF-β pathway genes in patients with RRMS.

## 1. Introduction

Multiple sclerosis (MS) is an inflammatory and demyelinating disease affecting the central nervous system (CNS). Its etiology is still unknown, although it is thought to be the result of immune dysregulation induced by both genetic and environmental factors [1]. The pathogenesis of this condition is mainly autoimmune, with the participation of both innate and acquired immunity. The immune response to CNS antigens is mediated by activated autoreactive CD4+ T cells, with contributions from other cells and immune system cytokines [2]. Of the various environmental factors related to the disease, vitamin D (VD) is thought to play a major role in development and progression owing to its role in immune regulation [3,4]. 

In studies of MS and in the murine animal model of experimental autoimmune encephalomyelitis (EAE), the presence of VD has been correlated with increased expression of anti-inflammatory cytokines (IL-4; IL-10) and transforming growth factor β (TGF-β), a decrease in the expression of certain pro-inflammatory cytokines such as IL-12, IL-17, and IFN-γ [5], and modulation of the T cell response [6], thus highlighting its immunoregulatory role. VD performs its functions by interacting with its receptor (VDR), which is expressed extensively on cells of the immune system. After VD activates the VDR, it binds to specific DNA regions, leading to promotion or suppression of gene transcription [7].

TGF-β is a superfamily of cytokines and growth factors involved in the processes of cell growth and differentiation in the immune system [8]. TGF-β signaling has important anti-inflammatory effects, such as inhibiting cell proliferation mediated by IL-2 and IL-12, decreasing the production of tumor necrosis factor (TNF) and lymphotoxins, and intervening in the maintenance of FOXP3+ regulatory T cells. However, in the presence of IL-6, it promotes differentiation toward Th17, a cell phenotype that plays an important role in the immunopathogenesis of EAE and MS [9,10,11].

Activation and phosphorylation of the TGF-β type II receptor (TGF-βRII) triggers R-SMAD proteins (SMAD2/3). These form a complex with the mediator Co-SMAD4, which translocates to the nucleus to regulate transcription of specific genes. Two inhibitory I-SMAD proteins (SMAD6 and SMAD7) run a negative feedback loop on TGF-β signaling, by competing with R-SMADs for receptor binding, thus inhibiting their phosphorylation [10,11]. Apart from the SMAD pathway, TGF-β signaling is carried out by other non–SMAD-dependent pathways (e.g., Ras-ERK, mitogen-activated protein kinase [MAPK], c-Jun N-terminal kinase [JNK], TAK-MKK3/6-p38) [12,13] and is modulated at several levels by regulatory proteins, such as activated STAT inhibitor proteins (PIAS), which are able to suppress SMAD3 transcriptional activity [14,15]. VD has been associated with the immunoregulatory functions of TGF-β. It has demonstrated the ability to increase transcription factors that bind to the TGF-β promoter, activate alternative pathways mediated by mitogen-activated protein kinase (MAPK), and modify the half-life of TGF-β mRNA, thus suggesting a post-transcriptional mechanism. The precise direct or indirect mechanisms by which VD interacts with TGF-β are still unclear [16].

The TGF-β superfamily also includes bone morphogenetic proteins (BMPs), which are a group of growth factors whose functioning also depends on signaling pathways common to TGF-β via SMAD- and non–SMAD-dependent pathways [17,18]. They have been linked to cell maintenance and differentiation processes in various tissues. In the CNS, they act primarily as promoters of astrocyte differentiation and suppressors of oligodendroglial differentiation [19]. They also play an important role in the immune system in T-cell differentiation and regulation of cellular and humoral responses [17,20]. BMP signaling has been shown to be altered in EAE and MS by promoting a pro-inflammatory response in the disease [21,22].

In this study, we aimed to analyze the regulatory effect of VD on TGF-β signaling pathway gene expression in vivo in patients with relapsing-remitting MS (RRMS). 

## 2. Results

### 2.1. Patient Characteristics

A total of 21 patients were included in the study. The median age of patients was 39 (IQR, 20–56) years, and most were women (76.2%). Natalizumab was the most common treatment (Table 1). Serum VD levels increased from baseline to six months.

### 2.2. Gene Expression of TGF-β Related Genes and Relationship with VD Supplementation

Analysis of TGF-β related gene expression revealed that the mean expression of the *SMAD7* gene increased 3.66-fold (*p* value = 0.033) from baseline to six months of treatment, and 5.32-fold (*p* value = 0.008) from the first to the sixth month of treatment. The mean expression of *SMAD7* in the first month was lower than at baseline, although the difference was not significant. Increased mean expression of *ERK1* was also observed at six months compared to one month of supplementation (1.84-fold; *p* value = 0.002). No significant differences were found in the mean expression of *ZMIZ1*, *BMP2*, *BMPRII*, *BMP4*, or *BMP5* at the study timepoints (Figure 1), although a trend toward overexpression of *BMP5* was observed at one month and six months of VD supplementation compared to baseline. No correlation was found between expression variables of the different study genes and serum VD levels. Individual expression data and variables used for comparison in all figures are provided in the Appendix A.

### 2.3. Gene Expression of TGF-β Related Genes under Natalizumab Treatment and VD Supplementation

As natalizumab was the main treatment, mean changes in TGF-β related gene expression were analyzed, specifically in this group of patients. The comparison revealed that only *BMPRII* was differentially expressed after one month of VD supplementation. Thus, *BMPRII* was overexpressed 3.66-fold in the non–natalizumab- versus natalizumab-treated group (*p* value = 0.001) (Table 1, Figure 2). In addition, a trend toward overexpression of *BMPRII* was observed in non–natalizumab-treated patients versus natalizumab-treated patients during follow-up (baseline, one month, and six months). However, it was only after one month that this difference became statistically significant. 

In contrast, there was a trend toward decreased mean expression of *BMP5* from baseline to six months in natalizumab-treated patients, while in non–natalizumab-treated patients, the trend was toward increased mean expression. Nevertheless, these changes were not statistically significant.

### 2.4. Gene Expression of TGF-β Related Genes and Clinical-Radiological Disease Activity 

The expression of genes related to inflammation could be associated with the clinical and radiological characteristics of affected patients. Therefore, we compared the group of patients whose Rio-Score improved or remained unchanged at one year of the study with those whose Rio-Score worsened and found a significant difference in the mean expression of *ERK1* at baseline (2.25-fold higher expression in the improvement or no change in Rio-Score group; *p* value = 0.045). Significant differences were also found in mean *BMP2* expression at baseline (7.44-fold higher expression in the worsening Rio-Score group; *p* value = 0.008) and after six months of supplementation (3.92-fold increased expression in the Rio-Score worsening group; *p* value = 0.023), and in mean *BMP5* expression at one month of supplementation (23-fold increased expression was observed in the worsening Rio-Score group; *p* value = 0.004) (Table 1, Figure 3). A trend toward increased expression of *BMPRII* after six months of supplementation was observed in the worsening group, and a trend toward increased expression of *SMAD7* was observed in both groups after six months of supplementation. However, these changes were not statistically significant.

A total of four patients experienced a relapse during the study. Comparison between patients who experienced a relapse and those who did not showed a significant difference in mean *BMP2* expression at baseline in the relapse group (5.94-fold; *p* value = 0.017) (Figure 4). A trend was also observed toward increased expression of *BMP5* at one month and *BMPRII* at one and six months in the group of patients who experienced a relapse. However, these changes were not statistically significant. 

Comparison between the patients whose EDSS improved at one year of the study and those whose disease worsened or remained unchanged showed a significant difference in *BMP5* mean expression at baseline (15.55-fold higher in the EDSS improvement group; *p* value = 0.019) (Table 1, Figure 5). In contrast, the trend in expression seems to have been lower at one and six months in the EDSS improvement group (data not statistically significant).

A non–statistically significant trend toward overexpression of *BMPRII* was also observed in the worsening or no-change EDSS group during follow-up (baseline, one month, and six months). 

## 3. Discussion

VD is an environmental factor that is intimately related to the development of MS [23]. It plays an immunoregulatory role by activating various pathways. One of these is the TGF-β pathway, which, through its different cytokines and growth factors, promotes mainly anti-inflammatory functions, such as the generation and maintenance of regulatory T cells or the inhibition of pro-inflammatory cytokines [24,25]. However, TGF-β is also able to promote differentiation toward Th17, a cell phenotype involved in the pathogenesis of MS and EAE [26,27]. To carry out these functions, TGF-β uses different signaling pathways. These may depend or not on SMAD proteins [28,29,30], which in turn are regulated at different levels by various regulatory proteins. Therefore, knowledge of the changes caused by VD supplementation in the expression of genes involved in this pathway is essential if we are to understand the molecular effects of this treatment and to establish future biomarkers of response to it. 

In this study, CD4+ T cells extracted from peripheral blood of RRMS patients, both before and after VD supplementation according to standard clinical practice, were examined for changes in the expression of TGF-β–related genes.

We found significant changes in *SMAD7* expression at six months from baseline and at one month from the start of supplementation. We also found significant differences in *ERK1* expression at six months compared to the first month after initiation of supplementation. SMAD7 exerts a negative feedback effect on the SMAD pathway of TGF-β [31,32], whereas ERK1 is part of the non-SMAD pathway [8,9,11]. These findings suggest that TGF-β is likely to carry out its functions primarily through the non-SMAD pathway, while the canonical SMAD pathway is inhibited due to overexpression of *SMAD7* under the influence of VD. An EAE study in mice revealed that when T cells lacked TGF-βRII, they did not differentiate to Th17, but when they were treated with a TGF-βRI kinase inhibitor (SB-431542) or overexpressed *Smad7*, the Th17 population was maintained through activation of non–SMAD-dependent genes [13]. We did not find an association between VD levels and the expression of these genes, thus suggesting that VD would be regulating them not directly, but in a more complex way, probably by acting on other, different pathways that indirectly end up favoring overexpression of *SMAD7* and *ERK1*. In this sense, other studies have demonstrated that VD can activate alternative non-SMAD pathways of TGF-β by activating transcription factors that bind to the TGF-β promoter or even influencing its post-transcriptional regulation [16]. However, in another study, again in mice, the authors found that the active form of VD (1,25(OH)_2_D3) promoted *Smad3* expression and inhibited *Smad7* expression during differentiation to Th17 [33]. In humans with MS supplemented with VD, one study found significant differences in expression in genes related to the Th17 population [34]. Another study, also in humans with MS who received VD, found increased expression of IL-10 but not TGF-β1 in treated patients [35]. The influence of VD on gene expression may change depending on the cell type of the immune system, as shown in another work in MS patients [36]. Despite the difference in study models, the mechanism by which VD regulates TGF-β is still unclear.

Th17 facilitates disruption of the blood–brain barrier, thus enabling the transit of inflammatory cells into the CNS [26]. IL-17A and other Th17-responsive cytokines have been detected in CNS lesions in both MS and EAE [37], although it is difficult to access the CNS compartment in MS [10]. Therefore, we investigated whether the mechanism of action of disease-modifying treatment (DMT) would influence TGF-β gene expression. In our study, all patients were receiving DMT, nine of them with natalizumab. Natalizumab is a monoclonal antibody directed against the α-chain of the VLA-4 integrin and is a very potent inhibitor of cell migration into tissues, including the CNS. It reduces relapses and active lesions very effectively in the MRI of RRMS patients [38,39]. Of all of the genes studied, significant differences in *BMPRII* expression were only found one month after initiation of supplementation in those patients who were not receiving natalizumab. A trend toward overexpression of *BMPRII* was observed in non–natalizumab-treated patients compared to natalizumab-treated patients, both at baseline and after six months. With respect to *BMP5*, we observed a decreasing trend in mean expression from baseline to six months in natalizumab-treated patients versus an increasing trend in non–natalizumab-treated patients. However, these changes were not statistically significant. The fact that there is an effective blockade of cell entry into the CNS depending on treatment with natalizumab and, therefore, apparently different CNS compartments, does not seem to be a determinant in modifying TGF-β gene expression, except in *BMPRII*. Nonetheless, it cannot be excluded that the other mechanisms of action of DMT will affect gene expression of TGF-β in regard to VD. This finding would be in line with other works, such as a study in which the authors measured expression of BMP genes in MS lesions and found increased expression of *BMP2*, *BMP4*, *BMP5*, *BMP7*, *BMPRII*, and *pSMAD1/5/8* in astrocytes, microglia/macrophages, and neurons [22]. In another study, in this case in EAE, overexpression of *Bmp4*, *Bmp6*, and *Bmp7* was observed in the lumbar spinal cord of mice with active EAE [40].

The Rio-Score is an index created from a combination of relapse data, sustained EDSS changes, and MRI lesions that is used to assess the degree of therapeutic response to interferon-beta in RRMS patients [41,42,43]. In our case, we used the Rio-Score as a simple measure of the clinical-radiological characteristics of the disease to assess whether they would influence gene expression in the study. Significantly higher expression of *ERK1* was found at baseline, normalizing at one month and six months after the initiation of supplementation in patients whose Rio-Score improved or remained unchanged at one year after the study. The higher expression of *ERK1* at baseline would suggest activation of the non-SMAD pathway of TGF-β [12,13], which normalized as soon as patients reached sufficiently high serum values of VD after one month of supplementation. Thus, the anti-inflammatory effects of TGF-β are mediated through the SMAD protein–dependent pathway. In patients with a worsening Rio-Score at one year, expression of *BMP2* was higher at baseline and six months, and that of *BMP5* was increased at one month, indicating that BMP signaling is altered in MS patients [17], although its expression does not seem to be related to VD levels and did not change after serum levels returned to normal. Along the same lines, we also observed overexpression of baseline *BMP2* in patients whose disease relapsed and overexpression of *BMP5* in patients whose EDSS improved, indicating dysregulation of BMP signaling in MS patients. One study found increased *BMP2* expression in the serum of untreated RRMS patients, and its levels correlated with those of *BMP4* and *BMP5* [44]. BMPs are multifunctional proteins involved in cell differentiation processes, such as astrogliogenesis, oligodendrogliogenesis, and neurogenesis [45,46,47,48]. BMP signaling also has effects on immune system differentiation, facilitating, together with TGF-β, the differentiation of FOXP3+ regulatory T cells [49]. In MS patients, this dysregulation of BMP signaling leads to proliferation of pro-inflammatory cell lines, thus promoting disease activity [21].

Given the lack of consensus on the dosage or the time of administration, VD supplementation was prescribed according to standard clinical practice criteria. In our study, there were no withdrawals from treatment or related adverse effects. This observation is in line with other published studies, which state that VD supplementation is safe and tolerable with immunological benefits in MS patients [6]. Data are currently available from two large clinical trials examining VD supplementation in MS (SOLAR and CHOLINE). While neither study met its primary endpoints (no evidence of disease activity [NEDA-3] at 48 weeks in SOLAR and annualized relapse rate at 96 weeks in CHOLINE), both showed effects on some secondary endpoints [50,51,52]. Although there are discrepancies between observational studies and clinical trials with VD, the evidence would still support the recommendation to prescribe supplementation in MS patients with insufficient VD levels [53,54].

Munger et al. studied changes in global gene expression in the blood cells of MS patients who received VD supplementation, suggesting that the activity of MS is affected by unbalanced VD-related gene expression [55]. The differences with our study are that, first, we focused on the changes in TGF-ß signaling, whereas Munger et al. provided a whole transcriptomic profile. Second, we measured expression in T lymphocytes, whereas Munger et al. used whole blood. And third, patients in the Munger study were treated with interferon ß-1b. However, our results also suggest that regulation of VD-expressed genes may affect the activity of MS.

A limitation of the study may be the lack of a healthy control group to measure gene expression changes after VD supplementation. The expression of *SMAD7* increased by a factor of 3.6 in multiple sclerosis (MS) patients after six months of VD supplementation. Similarly, *SMAD7* expression exhibited a 2.3-fold increase in healthy donors compared to MS patients [10]. While our study did not incorporate a healthy control group, our findings suggest a restoration of "normal" *SMAD7* levels after six months of VD supplementation [10]. No comparisons are available for the remaining genes analyzed in this study.

## 4. Materials and Methods

### 4.1. Patients

Patients were recruited and samples taken at the Neurology Department of Hospital General Universitario Gregorio Marañón, Madrid, Spain between 2016–2017. The inclusion criterion was a diagnosis of RRMS according to the 2010 McDonald criteria with or without DMT in patients with insufficient VD levels (<30 ng/mL). Patients received VD supplement (calcifediol 0.266 mg) at a dosage of 1 ampoule orally per month. The exclusion criteria included relapse or corticosteroid treatment in the previous 30 days and contraindication to VD supplementation. In Figure 1, a healthy control group supplemented with vitamin D (VD) was not included. This decision was based on our group’s previous data, where we observed initial differences in gene expression for SMAD7 between healthy donors and individuals with multiple sclerosis (MS) [10]. Therefore, including such differences could potentially obscure any effects of vitamin D on gene expression in healthy donors. Relapses were defined as new symptoms or worsening of pre-existing neurological symptoms lasting more than 24 h after a period of 30 days of improvement or stability in the absence of fever or infection. Patients signed a written informed consent form. This study followed the recommendations of the Declaration of Helsinki and was approved by the ethics committee of Hospital General Universitario Gregorio Marañón (234/16).

### 4.2. Isolation and Culture of CD4+ T Cells

Blood samples (20 mL) were drawn in EDTA anticoagulant tubes at baseline and 1 month and 6 months after initiation of VD supplementation. Samples were processed immediately after collection under sterile conditions. Peripheral blood mononuclear cells were isolated by Ficoll gradient as described in [56]. CD4+ T-lymphocyte subtypes were negatively selected using the DynaBeads^®^ UntouchedTM Human CD4 T Cells kit (Invitrogen, Waltham, MA, USA). Cells were counted, homogenized in Ribozol (Amresco, Solon, OH, USA), and stored at −80 °C in pellet form until RNA extraction.

### 4.3. RNA Isolation and cDNA Extraction

RNA from CD4+ T lymphocytes was isolated using the Ribozol Plus RNA purification kit (Amresco) following the manufacturer’s instructions. The concentration of the RNA obtained was quantified using spectrophotometry in a Quawell Q5000 device (Quawell Technology, Sunnyvale, CA, USA), and its integrity was verified in a 2100 Bioanalyzer device (Agilent Technologies, Santa Clara, CA, USA) using the RNA 6000 kit (Agilent). All RNAs had an integrity number (RIN) ≥ 8 and were, therefore, used for semi-quantitative real-time PCR (qRT-PCR).

### 4.4. Quantitative Real-Time PCR (qRT–PCR)

Candidate genes representative of the different TGF-β pathways were selected. *SMAD7* was selected as a regulatory gene for the canonical SMAD pathway, *ERK1* as a representative gene for the non-SMAD pathway, *ZMIZ1* as a regulatory gene for PIAS expression, and *BMPs* as members of the TGF-β superfamily. We selected *BMP2*, *BMPRII*, *BMP4*, and *BMP5* for their increased expression in the MS lesions assessed in previous studies. Changes in selected genes were analyzed in 21 RRMS patients at baseline and one month and six months after the start of VD supplementation. The cDNAs were generated from 500 ng of RNA using the High Capacity cDNA Archive Kit (Invitrogen) in 20 μL of final volume at 37 °C for 2 h. The enzyme was inactivated at 85 °C for 15 min. Relative quantification of gene expression was performed using qRT-PCRs in triplicate with 2 μL of cDNA per well (1/10 dilution), 1 × SYBR Green PCR Master Mix (Roche Applied Science, Penzberg, Germany), and 0.04 μM of oligonucleotides specific for *SMAD7* (F-ACCCGATGGATTTTCTCAA; R-AGGGGCCAGATAATTCGTTC), *ERK1* (F-CCCTAGCCCAGACAGACATC; R-GCACAGTGTCCATTTTCTAACAGT), *ZMIZ1* (F-TTAGAGGGTCAGGCCGGAGC; R-TCGGGAAGGAGATCCAGCGAA), *BMP2* (F-TCCACCATGAAGAATCTTTG; R-TAATTCGGTGATGGAAACTG), *BMPRII* (F-CCCGCTCCTACCTCTCCT; R-CGCAGAACAACCGTGAGAG), *BMP4* (F-CTGCAACCGTTCAGAGGTC; R-TGCTCGGGATGGCACTAC), *BMP5* (F-ATGGCAGGACTGGATTATAG; R-AGAGTCTGAACTATAGCGTG), and *GAPDH* (F-AGCCACATCGCTCAGACAC; R-GCCCAATACGACCAAATCC). PCR reactions were performed on a StepOne Plus thermal cycler (Applied Biosystems, Foster City, CA, USA). The relative expression of the genes analyzed was quantified using the 2^−∆∆Ct^ method with StepOnePlus v2.3 software (Applied Biosystems). The *GAPDH* gene was used to normalize expression.

### 4.5. Clinical-Radiological Disease Activity

Clinical-radiological disease activity was measured at baseline and one year after the start of supplementation using the Rio-Score, which combines relapse, EDSS, and magnetic resonance imaging (MRI) criteria. The relapse criterion scored one point if the patient had ≥1 relapse per year. The EDSS criterion scored one point if the patient had an increase of ≥1 point on the EDSS scale, maintained for at least 6 months and confirmed at 1 year of the study. The MRI criterion scored one point if the patient had >2 active lesions (with respect to a previous annual MRI), defined as new or T2-enhancing lesions, plus the number of gadolinium-enhancing T1 lesions. The Rio-Score value was between 0 and 3. An increase in the Rio-Score value at 1 year compared to baseline was considered a deterioration in the Rio-Score, whereas a decrease in the value at 1 year compared to baseline was considered an improvement in the Rio-Score. Alternatively, an increase in the EDSS score at 1 year compared to baseline was considered a deterioration in the EDSS, while a decrease in the score at 1 year compared to baseline was considered an improvement in the EDSS.

### 4.6. Statistical Analysis

For qRT-PCR, relative expression was analyzed using the 2^−ΔΔCt^ method with StepOne Software v2.3 (Applied Biosystems). Relative quantification was measured using Expression Suite software v1.3 (Life Technologies, Carlsbad, CA, USA). GraphPad Prism (v.8, Irvine, CA, USA) software was used for expression graphing. ANOVA and a paired *t* test were performed to identify differentially expressed genes. The Pearson correlation was used to study the association between variables. A *p* value ≤ 0.05 was considered statistically significant. The clinical-radiological activity variables were analyzed using IBM SPSS Statistics for Windows, Version 26.0 (IBM Corp., Armonk, NY, USA).

## 5. Conclusions

VD indirectly regulates the in vivo expression of genes related to the TGF-β signaling pathway in RRMS patients. Increased expression of *SMAD7* and *ERK1* suggests that VD could affect both SMAD and non-SMAD signaling pathways, which are critical for the dual functions of TGF-β in the immune system and in MS. Knowledge of the mechanisms of action of VD may yield biomarkers of response. Our study could have been limited by the impact of DMT mechanisms of action on gene expression and the evaluation of other subtypes of peripheral blood mononuclear cells. Further studies in larger cohorts are needed to elucidate the mechanisms by which VD exerts this regulation.

## Figures and Tables

**Figure 1 ijms-24-14447-f001:**
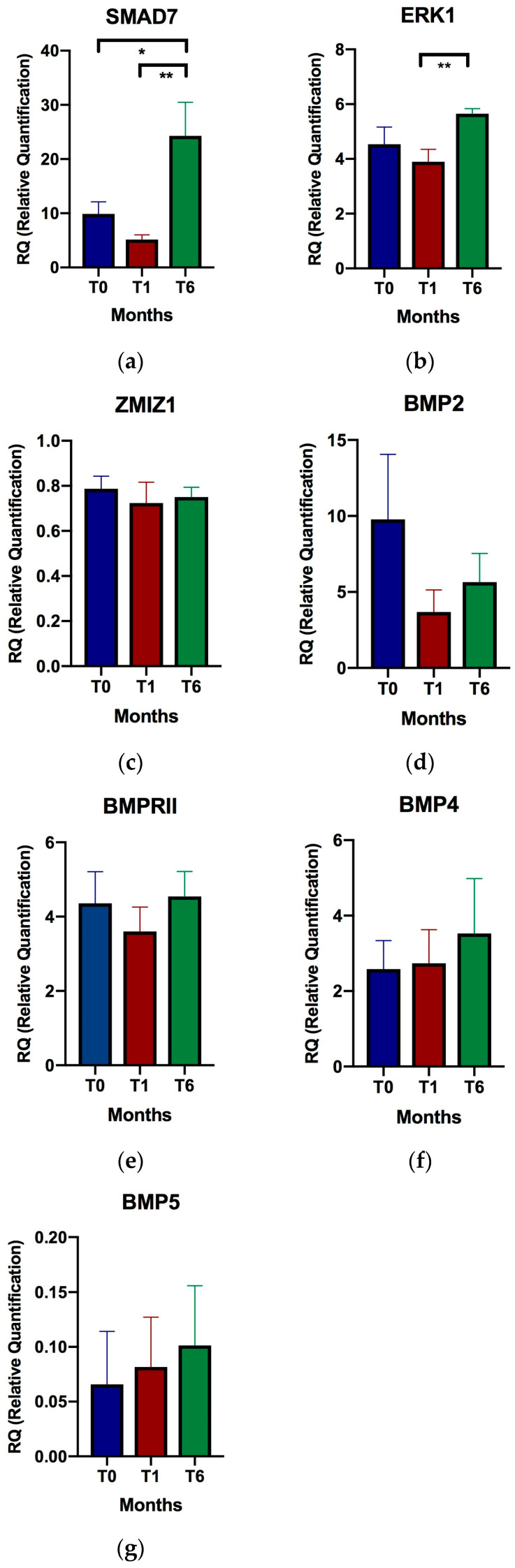
Relative expression of selected genes in relapsing-remitting multiple sclerosis patients at baseline, after one month of supplementation, and after six months of supplementation. (**a**) *SMAD7*; (**b**) *ERK1*; (**c**) *ZMIZ1*; (**d**) *BMP2*; (**e**) *BMPRII*; (**f**) *BMP4*; and (**g**) *BMP5*; T0, baseline; T1, after one month of supplementation; T6, after six months of supplementation. * *p* < 0.05; ** *p* < 0.01. Relative quantification of expression is represented as the mean of relative individual expression value; the error bars indicate the standard error of the mean.

**Figure 2 ijms-24-14447-f002:**
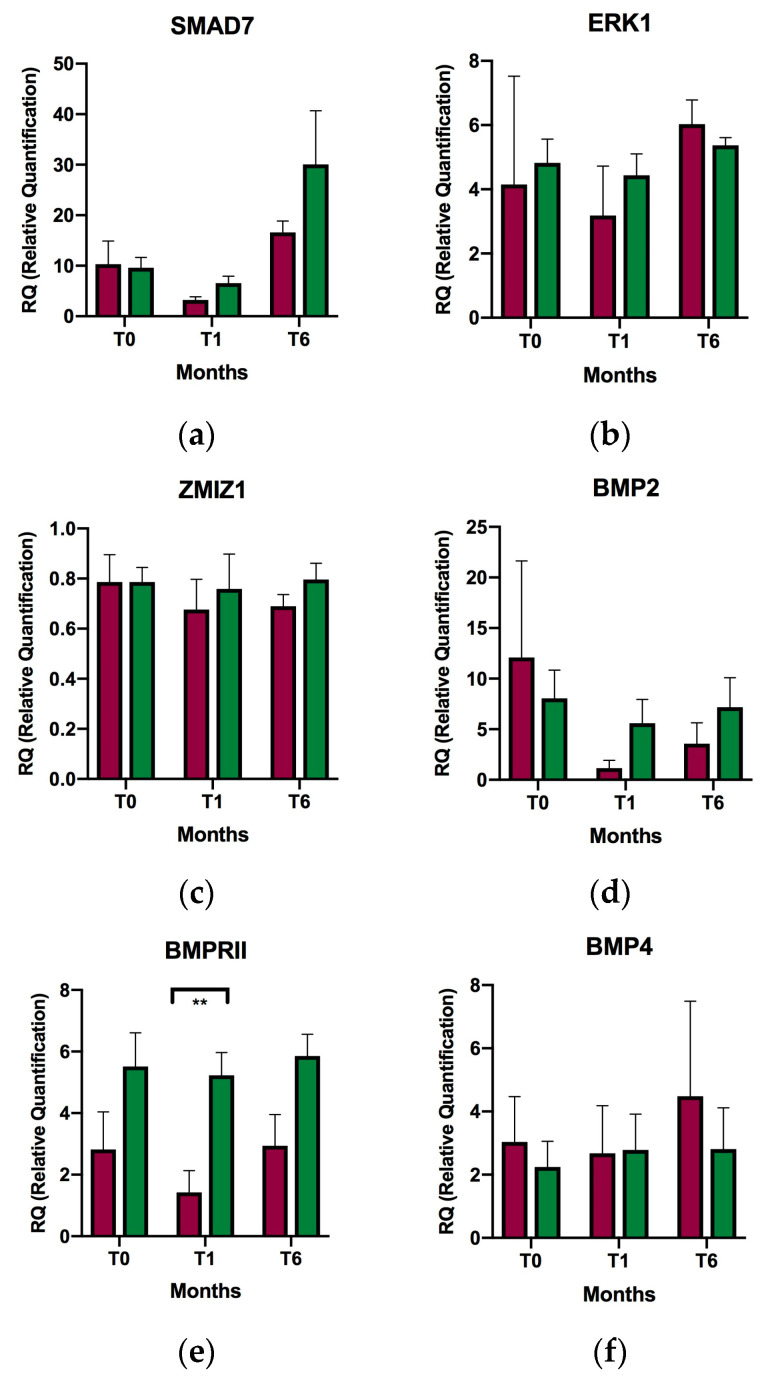
Relative expression of selected genes in relapsing-remitting multiple sclerosis patients with and without natalizumab at baseline, after one month of supplementation, and after six months of supplementation. (**a**) *SMAD7*; (**b**) *ERK1*; (**c**) *ZMIZ1*; (**d**) *BMP2*; (**e**) *BMPRII*; (**f**) *BMP4*; and (**g**) *BMP5*; T0, baseline; T1, after one month of supplementation; T6, after six months of supplementation; Red, natalizumab; Green, no natalizumab. ** *p* < 0.01. Relative quantification of expression is represented as the mean of relative individual expression values; the error bars indicate the standard error of the mean.

**Figure 3 ijms-24-14447-f003:**
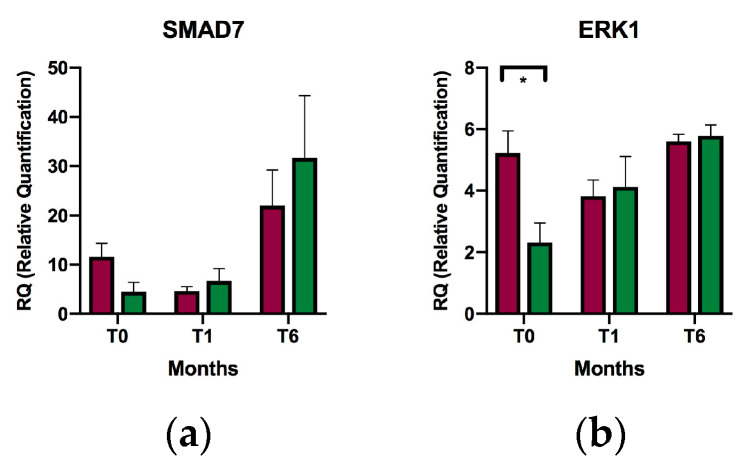
Relative expression of selected genes in relapsing-remitting multiple sclerosis patients with improvement or no change in the Rio-Score versus worsening in the Rio-Score at baseline, after one month of supplementation, and after six months of supplementation. (**a**) *SMAD7*; (**b**) *ERK1*; (**c**) *ZMIZ1*; (**d**) *BMP2*; (**e**) *BMPRII*; (**f**) *BMP4*; and (**g**) *BMP5*; T0, baseline; T1, after one month of supplementation; T6, after six months of supplementation; Red, improvement or no change in the Rio-Score; Green, worsening in the Rio-Score. * *p* < 0.05; ** *p* < 0.01. Relative quantification of expression is represented as the mean of relative individual expression values; the error bars indicate the standard error of the mean.

**Figure 4 ijms-24-14447-f004:**
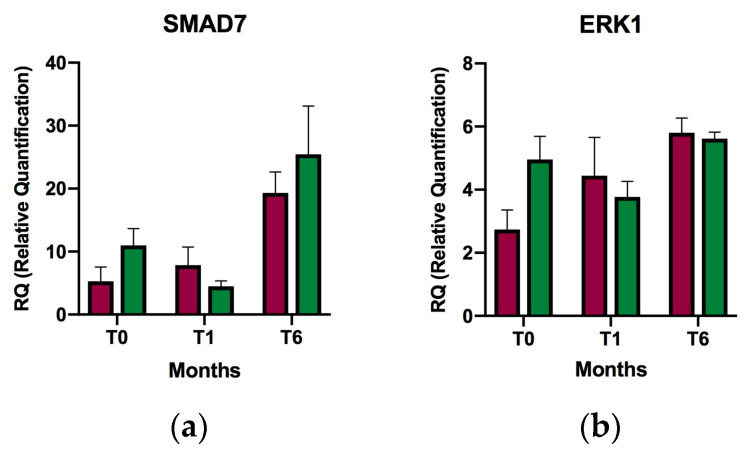
Relative expression of selected genes in relapsing-remitting multiple sclerosis patients with or without relapses of the disease at baseline, after one month of supplementation and after six months of supplementation. (**a**) *SMAD7*; (**b**) *ERK1*; (**c**) *ZMIZ1*; (**d**) *BMP2*; (**e**) *BMPRII*; (**f**) *BMP4*; and (**g**) *BMP5*; T0, baseline; T1, after one month of supplementation; T6, after six months of supplementation; Red, relapse; Green, no relapse. * *p* < 0.05. Relative quantification of expression is represented as the mean of relative individual expression values; the error bars indicate the standard error of the mean.

**Figure 5 ijms-24-14447-f005:**
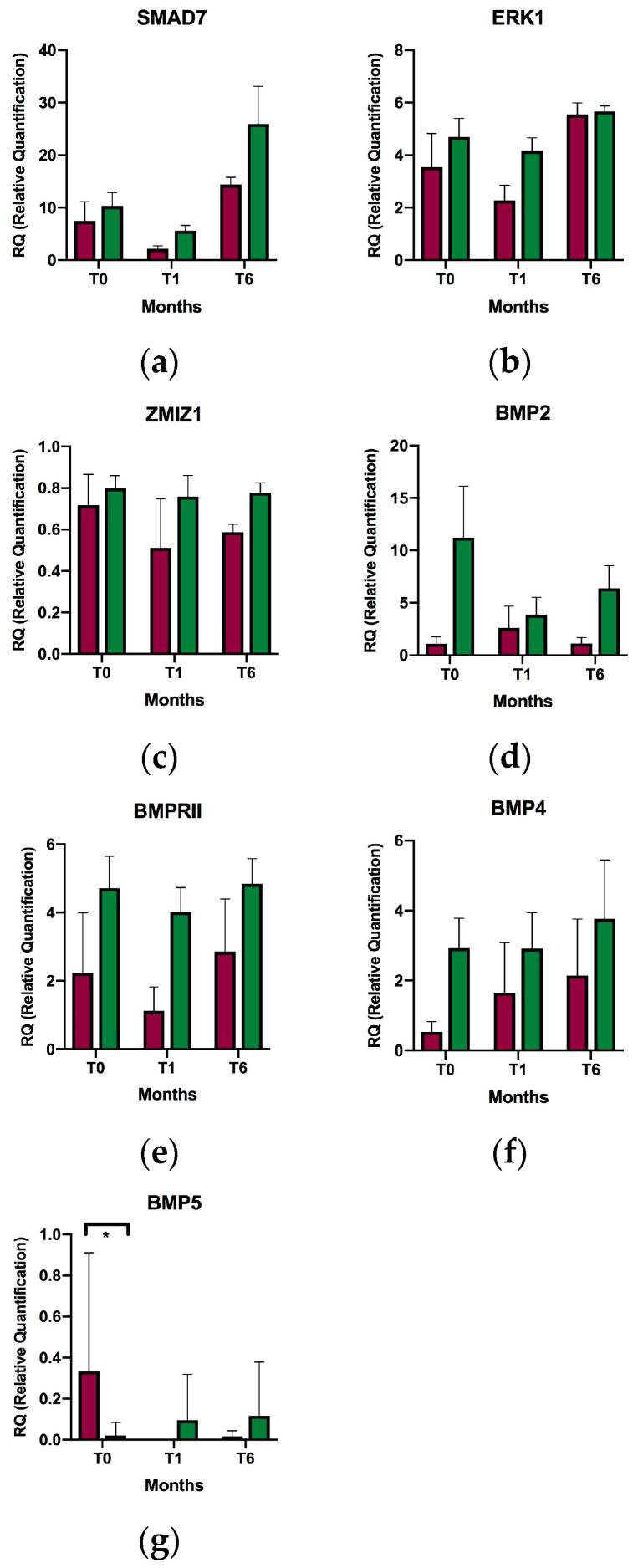
Relative expression of selected genes in relapsing-remitting multiple sclerosis patients with improvement in the EDSS versus worsening or no change in the EDSS at baseline, after one month of supplementation, and after six months of supplementation. (**a**) *SMAD7*; (**b**) *ERK1*; (**c**) *ZMIZ1*; (**d**) *BMP2*; (**e**) *BMPRII*; (**f**) *BMP4*; and (**g**) *BMP5*; T0, baseline; T1, after one month of supplementation; T6, after six months of supplementation; Red, improvement in the EDSS; Green, worsening or no change in the EDSS. * *p* < 0.05. Relative quantification of expression is represented as the mean of relative individual expression values; the error bars indicate the standard error of the mean.

**Table 1 ijms-24-14447-t001:** Patient characteristics.

Characteristic	RRMS (*n* = 21)
Median age, years (IQR, min–max)	39 (11.5, 20–56)
Sex	
Women *n*, (%)	16 (76.2%)
Men *n*, (%)	5 (23.8%)
Months from diagnosis to sample collection, Median (IQR, min–max)	95 (115.3, 15–272)
Type of disease-modifying treatment (*n*)	
Baseline–end	
Interferon beta-1a im	3–3
Glatiramer acetate	4–3
Teriflunomide	2–1
Dimethyl fumarate	3–3
Natalizumab	9–10
Alemtuzumab	0–1
Serum vitamin D levels (ng/mL) Median, IQR (min–max)	
T0	24.4, 5.5 (10.1–29.5)
T1	33.8, 11.2 (13.1–80.4)
T6	35.8, 6.9 (17.0–48.0)
EDSS Median, IQR (min–max) Baseline–end	1.0, 2.2 (0–6)
Rio-Score *n*, (%) Baseline–end	
0	10 (47.6%)–15 (71.4%)
1	7 (33.3%)–5 (23.8%)
2	2 (14.3%)–1 (4.8%)
3	1 (4.8%)–0 (0%)

RRMS, relapsing-remitting multiple sclerosis; im, intramuscular; T0, baseline; T1, after one month of supplementation; T6, after six months of supplementation; EDSS, expanded disability status scale.

## Data Availability

Data available in Appendix A.

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
