# Peer review of "Changes in the Expression of TGF-Beta Regulatory Pathway Genes Induced by Vitamin D in Patients with Relapsing-Remitting Multiple Sclerosis"

_ijms, 2023, doi:10.3390/ijms241914447_

Round 1

Reviewer 1 Report

Manuscript ID: ijms-2540912

Manuscript Title: Changes in the expression of TGF-beta regulatory pathway 2
genes induced by vitamin D in patients with relaps- 3 ing-remitting multiple sclerosis

Authors: Alberto Lozano-Ros, María L. Martínez-Ginés, José M. García-Domínguez, Sara Salvador-Martín, Haydee Goicochea-Briceño, Juan P. Cuello, Ariana Meldaña-Rivera, Yolanda Higueras-Hernández, María Sanjur jo-Sáez, Luis A. Álvarez-Sala-Walther and Luis A. López-Fernández

In this study, the authors analyzed the expression of genes associated with the TGF-B pathway individuals with relapsing-remitting multiple sclerosis who are supplemented with Vitamin D. The study is interesting and has some real-world medical implications. However, the manuscript requires substantial revision before it is acceptable for publication.
Below are a list of comments in no particular oder of significance.

1. In line 44-45, the authors begin the sentence with “As a result, vitamin D inhibits the expression of...”. Is “As a result” an accurate description? Do the authors mean that because vitamin D increases the expression of an- 43 ti-inflammatory cytokines (IL-4; IL-10) and transforming growth factor β (TGF-β), it inhibits expression of certain pro-inflammatory cytokines? or is this just correlated?

2. In the Introductory section, it would be great if the authors could include a paragraph justifying why they chose the specific genes that they did. After all, they discuss other genes such as SMAD2/3, however, those are not analyzed.

3. The section headings in the Results section are very vague. The authors should rewrite them in a more straightforward manner. For example: In “2.2. Gene expression of TGF-β related genes and relationship with vitamin D”, instead of “relationship with vitamin D”, the authors could just say something like “with Vitamin D supplementation”. Same can be said for section 2.3. Instead of “2.3. Gene expression of TGF-β–related genes and disease-modifying treatment” The authors could just say “2.3. Gene expression of TGF-β–related genes under natalizumab treatment and vitamin D supplementation” or something along these lines. After all, it is only natalizumab treatment that is analyzed, no reason to generalize to “disease-modifying treatment”.

4. In the beginning of section 2.2, the authors write: “Analysis of TGF-β–related gene expression revealed that expression of the SMAD7 gene increased 3.66-fold (p value=0.033)…” Do they mean average expression? if so, it would be good to state that.

5. Also in the beginning of section 2.2, the authors write: “Analysis of TGF-β–related gene expression revealed that expression of the SMAD7 gene increased 3.66-fold (p value=0.033) from baseline to 6 months of treatment, and 5.32-fold (p value=0.008) from the first to sixth month of treatment.“ Does this mean that expression went down between T0 and T1, then rose for T6? How could it be only 3.66 between T0 - T6 but 5.32 between T1 and T6? How come this anomaly isn’t mentioned in the paper?

6. If BMP7 was not amplified and was not studied, it should not be included in the manuscript at all, including in the Abstract.

7. Line 100, the authors use a new abbreviation “VD”, without explaining what it is. Abbreviations should be explained at their first use in the paper and continue to eb used regularly. It can get very confusing for the reader if the authors vacillate between abbreviation and writing the words out.

8. A small color-coding key box within the figure for the graphs in Figures 2, 3, 4, and 5 would be great.  The authors should also consider using different colors for the various graphs.

9. In Figure 2 especially, but also in the rest of the bar graphs, error bars are quite large. Have the authors considered removing outliers before statistical analysis? It appears from Figure 1 that there may be some of those which are disruptive.

10. In Figure 2 , BMPRII-T6 and BMP5-T6 appear to show very large differences but they have not been shown to be significantly different. This is also the case in Figures 3, 4, and 5 where there are what appear to be very large differences in the graphs, especially for  BMPRII-T6 and BMP5-T6, but they were not calculated to be significant and are not discussed by there authors. Have the authors looked closely at this?

11. For section 2.4, the authors state that the baseline levels show “increased expression” or decrease expression, but baseline is not going up or down, it is just higher or lower. Therefore these terms should likely be replaced with just higher or lower expression.

12. Is there a reason why the authors used a dot plot for the first graph set in Figure 1 but then changed to bar graphs for the rest?

13. Line 188, the authors introduce the abbreviation  “disease-modifying treatment (DMT)”. This should be used earlier in the paper and used consistently. 

14. In lines 189-190, the authors state “In our study, all patients were receiving DMT, nine of them with natalizumab.” Could it be that the different treatments that patients received affected gene expression differently? affecting the results? hence the outliers and the large error bars? This is something that should at least be discussed by the authors.

Author Response

Reviewer 1

In this study, the authors analyzed the expression of genes associated with the TGF-B pathway individuals with relapsing-remitting multiple sclerosis who are supplemented with Vitamin D. The study is interesting and has some real-world medical implications. However, the manuscript requires substantial revision before it is acceptable for publication.
Below are a list of comments in no particular order of significance.

1. In line 44-45, the authors begin the sentence with “As a result, vitamin D inhibits the expression of...”. Is “As a result” an accurate description? Do the authors mean that because vitamin D increases the expression of anti-inflammatory cytokines (IL-4; IL-10) and transforming growth factor β (TGF-β), it inhibits expression of certain pro-inflammatory cytokines? or is this just correlated?

The expression has been modified to ensure a more accurate description, as follows:

In studies of MS and in the murine animal model of experimental autoimmune encephalomyelitis (EAE), the presence of VD has been correlated with increased expression of anti-inflammatory cytokines (IL-4; IL-10) and transforming growth factor β (TGF-β), a decrease in the expression of certain pro-inflammatory cytokines such as IL-12, IL-17, and IFN-γ [5], and modulation of the Th cell response [6], thus highlighting its immunoregulatory role. VD performs its functions by interacting with its receptor (VDR), which is expressed extensively on cells of the immune system. After VD activates the VDR, it binds to specific DNA regions, leading to promotion or suppression of gene transcription [7].”.

2. In the Introductory section, it would be great if the authors could include a paragraph justifying why they chose the specific genes that they did. After all, they discuss other genes such as SMAD2/3, however, those are not analyzed.

We have added a paragraph in the Material and Methods section to justify the genes selected. We believe that Materials and Methods is a more appropriate section to address the choice of genes.

The text has been modified, as follows:

“Candidate genes representative of the different TGF-β pathways were selected. SMAD7 was selected as a regulatory gene for the canonical SMAD pathway, ERK1 as a representative gene for the non-SMAD pathway, ZMIZ1 as a regulatory gene for PIAS expression, and BMPs as members of the TGF-β superfamily. We selected BMP2, BMPRII, BMP4, and BMP5 for their increased expression in the MS lesions assessed in previous studies.”

3. The section headings in the Results section are very vague. The authors should rewrite them in a more straightforward manner. For example: In “2.2. Gene expression of TGF-β related genes and relationship with vitamin D”, instead of “relationship with vitamin D”, the authors could just say something like “with Vitamin D supplementation”. Same can be said for section 2.3. Instead of “2.3. Gene expression of TGF-β–related genes and disease-modifying treatment” The authors could just say “2.3. Gene expression of TGF-β–related genes under natalizumab treatment and vitamin D supplementation” or something along these lines. After all, it is only natalizumab treatment that is analyzed, no reason to generalize to “disease-modifying treatment”.

We have rewritten the titles based on the proposed suggestions to make them clearer in the manuscript.

4. In the beginning of section 2.2, the authors write: “Analysis of TGF-β–related gene expression revealed that expression of the SMAD7 gene increased 3.66-fold (p value=0.033)…” Do they mean average expression? if so, it would be good to state that.

The gene expression values are the mean of individual relative values of expression for each group compared. We have added the word “mean” to the text. We have also added information in the figures to explain that relative expression is represented as mean expression, with the error bars corresponding to the standard error of the mean.

5. Also in the beginning of section 2.2, the authors write: “Analysis of TGF-β–related gene expression revealed that expression of the SMAD7 gene increased 3.66-fold (p value=0.033) from baseline to 6 months of treatment, and 5.32-fold (p value=0.008) from the first to sixth month of treatment.“ Does this mean that expression went down between T0 and T1, then rose for T6? How could it be only 3.66 between T0 - T6 but 5.32 between T1 and T6? How come this anomaly isn’t mentioned in the paper?

While the mean expression of SMAD7 at T1 was lower than at T0, this change was not statistically significant. Thus, it was not considered a change and was not mentioned in the manuscript. At T6, mean expression of SMAD7 increased significantly vs T0. When comparisons were made using the t test, significant values were found for SMAD7 expression between T0 and T6 and highly significant values were found between T1 and T6.

We have modified the sentence to clarify this point, as follows:

“Analysis of TGF-β–related gene expression revealed that the mean expression of the SMAD7 gene increased 3.66-fold (p value=0.033) from baseline to six months of treatment, and 5.32-fold (p value=0.008) from the first to the sixth month of treatment. The mean expression of SMAD7 in the first month was lower than at baseline, although the difference was not significant.”

6. If BMP7 was not amplified and was not studied, it should not be included in the manuscript at all, including in the Abstract.

We have eliminated BMP7 from the manuscript.

7. Line 100, the authors use a new abbreviation “VD”, without explaining what it is. Abbreviations should be explained at their first use in the paper and continue to eb used regularly. It can get very confusing for the reader if the authors vacillate between abbreviation and writing the words out.

The abbreviation has been expanded at first use.

8. A small color-coding key box within the figure for the graphs in Figures 2, 3, 4, and 5 would be great.  The authors should also consider using different colors for the various graphs.

Since the use of different colors in each figure may prove confusing, we have decided to follow the same color scheme in all of them. In a previous version of the manuscript, we included a small box inside the figure with the color code key. The journal template, however, made the graphs too small, thereby increasing the difficulty in observing the results. The color code used in the figures is elucidated in each of them, and the reader needs only to refer to these figures. If the reviewer would like us to change the colors of the graphs, we will do so. However, we do not believe it is necessary for the reasons stated.

9. In Figure 2 especially, but also in the rest of the bar graphs, error bars are quite large. Have the authors considered removing outliers before statistical analysis? It appears from Figure 1 that there may be some of those which are disruptive.

The individual data were revised and two outliers were removed. This had a significant impact on reducing the size of the error bars. In addition, the error bars now indicate the standard error of the mean, which is remarkably small. These changes do not affect the previous statistically significant data.

  1. In Figure 2, BMPRII-T6 and BMP5-T6 appear to show very large differences but they have not been shown to be significantly different. This is also the case in Figures 3, 4, and 5 where there are what appear to be very large differences in the graphs, especially for  BMPRII-T6 and BMP5-T6, but they were not calculated to be significant and are not discussed by there authors. Have the authors looked closely at this?

Initially, we did not include the text for results that were not statistically significant. In this version of the manuscript, we have added comments for the BMPRII-T6 and BMP5-T6 results in all comparisons, as suggested by the reviewer.

Two new sentences have been added to the manuscript, as follows:

“In addition, a trend toward overexpression of BMPRII was observed in non–natalizumab-treated patients versus natalizumab-treated patients during follow-up (baseline, one month, and six months). However, it was only after one month that this difference became statistically significant. 

In contrast, there was a trend toward decreased mean expression of BMP5 from baseline to six months in natalizumab-treated patients, while non–natalizumab-treated patients, the trend was toward increased mean expression. Nevertheless, these changes were no statistically significant.”

  1. For section 2.4, the authors state that the baseline levels show “increased expression” or decrease expression, but baseline is not going up or down, it is just higher or lower. Therefore these terms should likely be replaced with just higher or lower expression.

The terms have been corrected as suggested, and the text now reads as follows:

“……..found a significant difference in the mean expression of ERK1 at baseline (2.25-fold higher expression in the improvement or no change Rio-Score group; p value=0.045). Significant differences were also found in mean BMP2 expression at baseline (7.44-fold higher ex-pression in the worsening Rio-Score …….”

12. Is there a reason why the authors used a dot plot for the first graph set in Figure 1 but then changed to bar graphs for the rest?

We have replaced the dot plot in figure 1 with bar graphs to ensure the consistency with the other figures in the manuscript.

13. Line 188, the authors introduce the abbreviation  “disease-modifying treatment (DMT)”. This should be used earlier in the paper and used consistently. 

The abbreviation has been introduced as requested.

14. In lines 189-190, the authors state “In our study, all patients were receiving DMT, nine of them with natalizumab.” Could it be that the different treatments that patients received affected gene expression differently? affecting the results? hence the outliers and the large error bars? This is something that should at least be discussed by the authors.

We compared patients who were receiving natalizumab with those who were not owing to the hypothetical difference in compartments (blood and CNS). Except for BMPRII at T1, we found no significant differences in either group but cannot exclude that other, different mechanisms of action may influence gene expression and explain tendencies or dispersions in the data. The outliers and the error bars are addressed above (point 9).

We have modified the Discussion section, as follows:

“In our study, all patients were receiving DMT, nine of them with natalizumab. Natalizumab is a monoclonal antibody directed against the α-chain of the VLA-4 integrin and is a very potent inhibitor of cell migration into tissues, including the CNS. It reduces relapses and active lesions very effectively in the MRI of RRMS patients [38,39]. Of all the genes studied, significant differences in BMPRII expression were only found one month after initiation of supplementation in those patients who were not receiving natalizumab. A trend toward overexpression of BMPRII was observed in non–natalizumab-treated patients compared to natalizumab-treated patients, both at baseline and after six months. With respect to BMP5, we observed a decreasing trend in mean expression from baseline to six months in natalizumab-treated patients versus an increasing trend in non–natalizumab-treated patients. However, these changes were not statistically significant. The fact that there is an effective blockade of cell entry into the CNS depending on treatment with natalizumab and, therefore, apparently different CNS compartments, does not seem to be determinant in modifying TGF-β gene expression, except in BMPRII. Nonetheless, it cannot be excluded that the other mechanisms of action of DMT will affect gene expression of TGF-β in regards to VD.”

Reviewer 2 Report

The manuscript titled "Changes in the expression of TGF beta regulatory pathway 2 genes induced by vitamin D in patients with relapsing remitting multiple sclerosis" presents an investigation into the regulatory effect of vitamin D on the TGF β signaling pathway in patients with relapsing remitting multiple sclerosis (RRMS). The study is of interest, as it explores the potential relationship between vitamin D and immune regulation in the context of RRMS. However, there are several areas that require significant improvement, including the introduction, conclusion, and overall language quality. Below, I provide specific comments for major modifications that should be made to enhance the manuscript's clarity, coherence, and academic rigor.

1. The introduction is insufficient in providing context and justification for the study. It is important to expand the introduction to thoroughly discuss relevant previous studies and establish a clear rationale for the investigation. Consider addressing the following points:

  • Introduce the significance of immune dysregulation in the pathogenesis of RRMS.
  • Provide a comprehensive overview of vitamin D signaling pathway e.g., https://doi.org/10.3390/ph14121222
  • Provide a comprehensive overview of the TGF β signaling pathway and its role in immune modulation.
  • Discuss the existing literature on the potential interplay between vitamin D and the TGF β pathway in autoimmune diseases, highlighting gaps in knowledge.
  • Clearly state the research objectives and hypotheses to guide readers through the study's focus.

2. The methodology section lacks crucial details about the study design, patient selection criteria, and experimental procedures. Provide the following information:

  • Clearly describe the inclusion and exclusion criteria for patient recruitment, ensuring that the RRMS diagnosis criteria are well-defined.
  • Specify the rationale behind selecting the specific genes (SMAD7, ERK1, ZMIZ1, BMP2, BMPRII, BMP4, BMP5, BMP7) for analysis and the methods used for their measurement.
  • Detail the disease modifying treatments administered to patients and explain how these treatments were integrated into the analysis.
  • Provide a clear explanation of how the Rio Score and EDSS were used to assess disease activity and progression.

3. The presentation of results is generally clear; however, the discussion lacks depth and coherence. Consider the following points for improvement:

  • Provide a comprehensive interpretation of the results, explaining the implications of changes in gene expression observed after vitamin D supplementation.
  • Discuss the potential mechanisms underlying the observed changes in SMAD7 and ERK1 expression, considering their roles in immune regulation.
  • Relate the findings to the broader context of RRMS and immune dysfunction, connecting the results to existing literature.
  • Address the lack of correlation between serum vitamin D levels and gene expression changes, discussing potential reasons for this discrepancy.
  • Clearly explain the significance of the changes in BMPRII, ERK1, BMP2, and BMP5 expression in relation to disease activity and treatment.

4. The conclusion is inadequately presented and does not effectively summarize the study's findings or their implications. Revise the conclusion to:

  • Succinctly recapitulate the main findings of the study, emphasizing the regulatory effect of vitamin D on specific genes in the TGF β pathway.
  • Discuss the potential clinical implications of these findings for RRMS management and treatment strategies.
  • Highlight the broader significance of the study in advancing our understanding of the complex interactions between vitamin D, immune regulation, and RRMS pathogenesis.
  • Clearly state the study's limitations and propose avenues for future research, such as exploring the molecular mechanisms underlying the observed gene expression changes.

5. The manuscript requires extensive language editing to improve its academic style, coherence, and clarity. Pay attention to sentence structure, grammar, punctuation, and appropriate use of terminology throughout the manuscript.

In summary, the manuscript presents a promising investigation into the regulatory effect of vitamin D on the TGF β pathway in RRMS patients. However, substantial revisions are required to enhance the manuscript's quality. Addressing the aforementioned comments related to introduction, methodology, results and discussion, conclusion, and language editing will greatly improve the overall presentation and impact of the study.

The manuscript requires extensive language editing to improve its academic style, coherence, and clarity. Pay attention to sentence structure, grammar, punctuation, and appropriate use of terminology throughout the manuscript.

Author Response

Reviewer 2

The manuscript titled "Changes in the expression of TGF beta regulatory pathway 2 genes induced by vitamin D in patients with relapsing remitting multiple sclerosis" presents an investigation into the regulatory effect of vitamin D on the TGF β signaling pathway in patients with relapsing remitting multiple sclerosis (RRMS). The study is of interest, as it explores the potential relationship between vitamin D and immune regulation in the context of RRMS. However, there are several areas that require significant improvement, including the introduction, conclusion, and overall language quality. Below, I provide specific comments for major modifications that should be made to enhance the manuscript's clarity, coherence, and academic rigor.

  1. The introduction is insufficient in providing context and justification for the study. It is important to expand the introduction to thoroughly discuss relevant previous studies and establish a clear rationale for the investigation. Consider addressing the following points:
  • Introduce the significance of immune dysregulation in the pathogenesis of RRMS.
  • Provide a comprehensive overview of vitamin D signaling pathway e.g., https://doi.org/10.3390/ph14121222
  • Provide a comprehensive overview of the TGF β signaling pathway and its role in immune modulation.
  • Discuss the existing literature on the potential interplay between vitamin D and the TGF β pathway in autoimmune diseases, highlighting gaps in knowledge.
  • Clearly state the research objectives and hypotheses to guide readers through the study's focus.

The various points within the section have been expanded. In addition, we have provided more detail for concepts such as dysregulation of the immune system in MS, the interaction between vitamin D and TGF-beta, and the objectives of the study. We have added more references to support the expansion of this section.

  1. The methodology section lacks crucial details about the study design, patient selection criteria, and experimental procedures. Provide the following information:
  • Clearly describe the inclusion and exclusion criteria for patient recruitment, ensuring that the RRMS diagnosis criteria are well-defined.
  • Specify the rationale behind selecting the specific genes (SMAD7, ERK1, ZMIZ1, BMP2, BMPRII, BMP4, BMP5, BMP7) for analysis and the methods used for their measurement.
  • Detail the disease modifying treatments administered to patients and explain how these treatments were integrated into the analysis.
  • Provide a clear explanation of how the Rio Score and EDSS were used to assess disease activity and progression.

We have provided a clearer explanation for the selection of patients, with emphasis on the RRMS criteria, as well as on the selection of candidate genes for analysis.

Table 1 (baseline and end of study) includes the list of DMTs. Considering the differences in hypothetical compartments (blood and CNS) between those who take and those who do not take this drug, we only considered natalizumab.

We have included an explanation of how the Rio-Score and the EDSS have been integrated into the results.

Concerning the rationale behind selecting the specific genes, we have included a new sentence:

“Candidate genes representative of the different TGF-β pathways were selected. SMAD7 was selected as a regulatory gene for the canonical SMAD pathway, ERK1 as a representative gene for the non-SMAD pathway, ZMIZ1 as a regulatory gene for PIAS expression, and BMPs as members of the TGF-β superfamily. We selected BMP2, BMPRII, BMP4, and BMP5 for their increased expression in the MS lesions assessed in previous studies.”

  1. The presentation of results is generally clear; however, the discussion lacks depth and coherence. Consider the following points for improvement:
  • Provide a comprehensive interpretation of the results, explaining the implications of changes in gene expression observed after vitamin D supplementation.
  • Discuss the potential mechanisms underlying the observed changes in SMAD7 and ERK1 expression, considering their roles in immune regulation.
  • Relate the findings to the broader context of RRMS and immune dysfunction, connecting the results to existing literature.
  • Address the lack of correlation between serum vitamin D levels and gene expression changes, discussing potential reasons for this discrepancy.
  • Clearly explain the significance of the changes in BMPRII, ERK1, BMP2, and BMP5 expression in relation to disease activity and treatment.

We have taken into account the reviewer’s suggestions. Our explanations have been improved, and we have provided a more detailed interpretation of the results. The explanation of the effects of SMAD7, ERK1, and BMPs on the immune system and their relationship with vitamin D have been improved.

  1. The conclusion is inadequately presented and does not effectively summarize the study's findings or their implications. Revise the conclusion to:
  • Succinctly recapitulate the main findings of the study, emphasizing the regulatory effect of vitamin D on specific genes in the TGF β
  • Discuss the potential clinical implications of these findings for RRMS management and treatment strategies.
  • Highlight the broader significance of the study in advancing our understanding of the complex interactions between vitamin D, immune regulation, and RRMS pathogenesis.
  • Clearly state the study's limitations and propose avenues for future research, such as exploring the molecular mechanisms underlying the observed gene expression changes.

We have rewritten the Conclusion to include the reviewer’s suggestions. The Discussion is now broader, including the significance of the findings, the relationship between vitamin D and immune dysregulation and MS, and the limitations.

  1. The manuscript requires extensive language editing to improve its academic style, coherence, and clarity. Pay attention to sentence structure, grammar, punctuation, and appropriate use of terminology throughout the manuscript.

The manuscript has been reviewed by a native-speaking medical writer.

In summary, the manuscript presents a promising investigation into the regulatory effect of vitamin D on the TGF β pathway in RRMS patients. However, substantial revisions are required to enhance the manuscript's quality. Addressing the aforementioned comments related to introduction, methodology, results and discussion, conclusion, and language editing will greatly improve the overall presentation and impact of the study.

Round 2

Reviewer 2 Report

Indeed, the authors have not addressed most of the raised concerns by the reviewer. Although, the authors stated that they have modified the manuscript, I can not find the changes that have been made??? The authors requested to highlight the modification made in the revised manuscript. Further, the conclusion still very short and not expressing the actual findings. Please again carefully check the concerns and provide detailed response:

3. The presentation of results is generally clear; however, the discussion lacks depth and coherence. Consider the following points for improvement:

·       Provide a comprehensive interpretation of the results, explaining the implications of changes in gene expression observed after vitamin D supplementation.

·       Discuss the potential mechanisms underlying the observed changes in SMAD7 and ERK1 expression, considering their roles in immune regulation.

·       Relate the findings to the broader context of RRMS and immune dysfunction, connecting the results to existing literature.

·       Address the lack of correlation between serum vitamin D levels and gene expression changes, discussing potential reasons for this discrepancy.

·       Clearly explain the significance of the changes in BMPRII, ERK1, BMP2, and BMP5 expression in relation to disease activity and treatment.

4. The conclusion is inadequately presented and does not effectively summarize the study's findings or their implications. Revise the conclusion to:

·       Succinctly recapitulate the main findings of the study, emphasizing the regulatory effect of vitamin D on specific genes in the TGF β pathway.

·       Discuss the potential clinical implications of these findings for RRMS management and treatment strategies.

·       Highlight the broader significance of the study in advancing our understanding of the complex interactions between vitamin D, immune regulation, and RRMS pathogenesis.

·       Clearly state the study's limitations and propose avenues for future research, such as exploring the molecular mechanisms underlying the observed gene expression changes.

5. The manuscript requires extensive language editing to improve its academic style, coherence, and clarity. Pay attention to sentence structure, grammar, punctuation, and appropriate use of terminology throughout the manuscript.

Although, the authors stated the manuscript has been revised by native speaker, the manuscript still contains merit of grammatical mistakes.

Author Response

Indeed, the authors have not addressed most of the raised concerns by the reviewer. Although, the authors stated that they have modified the manuscript, I can not find the changes that have been made??? The authors requested to highlight the modification made in the revised manuscript. Further, the conclusion still very short and not expressing the actual findings. Please again carefully check the concerns and provide detailed response:

We reproduce, point by point, the questions raised by the reviewer as they appeared in the last version of the manuscript. The exact sentence or sentences that address each question are now as they appeared in the manuscript. We do not know whether the reviewer finds these explanations inadequate or insufficient, but they are in the manuscript. This manuscript describes the changes in gene expression in CD4+ T lymphocytes of essential genes involved in TGF-beta signaling in RRMS treated with VD. This is a major limitation in the interpretation of the results as mentioned in the manuscript. The manuscript has undergone a second round of review by a native-speaking medical writer with extensive experience in the field. Only a few bugs were found and fixed.

  1. The presentation of results is generally clear; however, the discussion lacks depth and coherence. Consider the following points for improvement:
  • Provide a comprehensive interpretation of the results, explaining the implications of changes in gene expression observed after vitamin D supplementation.

“We found significant changes in SMAD7 expression at six months from baseline and at one month from the start of supplementation. We also found significant differences in ERK1 expression at six months compared to the first month after initiation of supplementation. SMAD7 exerts a negative feedback effect on the SMAD pathway of TGF-β [31,32], whereas ERK1 is part of the non-SMAD pathway [8,9,11]. These findings suggest that TGF-β is likely to carry out its functions primarily through the non-SMAD pathway, while the canonical SMAD pathway is inhibited due to overexpression of SMAD7 under the influence of VD.”

  • Discuss the potential mechanisms underlying the observed changes in SMAD7 and ERK1 expression, considering their roles in immune regulation.

“We did not find an association between VD levels and the expression of these genes, thus suggesting that VD would be regulating them not directly, but in a more complex way, probably by acting on other, different pathways that indirectly end up favoring overexpression of SMAD7 and ERK1. In this sense, other studies have demonstrated that VD can activate alternative non-SMAD pathways of TGF-β by activating transcription factors that bind to the TGF-β promoter or even influencing its post-transcriptional regulation [16].”

  • Relate the findings to the broader context of RRMS and immune dysfunction, connecting the results to existing literature.

“An EAE study in mice revealed that when T cells lacked TGF-βRII, they did not differentiate to Th17, but when they were treated with a TGF-βRI kinase inhibitor (SB-431542) or overexpressed Smad7, the Th17 population was maintained through activation of non–SMAD-dependent genes [13].”

“However, in another study, again in mice, the authors found that the active form of VD (1,25(OH)2D3) promoted Smad3 expression and inhibited Smad7 expression during differentiation to Th17 [33]. In humans with MS supplemented with VD, one study found significant differences in expression in genes related to the Th17 population [34]. Another study, also in humans with MS who received VD, found increased expression of IL-10 but not TGF-β1 in treated patients [35]. The influence of VD on gene expression may change depending on the cell type of the immune system, as shown in another work in MS patients [36]. Despite the difference in study models, the mechanism by which VD regulates TGF-β is still unclear.”

  • Address the lack of correlation between serum vitamin D levels and gene expression changes, discussing potential reasons for this discrepancy.

“In this sense, other studies have demonstrated that VD can activate alternative non-SMAD pathways of TGF-β by activating transcription factors that bind to the TGF-β promoter or even influencing its post-transcriptional regulation [16].”

“The influence of VD on gene expression may change depending on the cell type of the immune system, as shown in another work in MS patients [36].”

  • Clearly explain the significance of the changes in BMPRII, ERK1, BMP2, and BMP5 expression in relation to disease activity and treatment.

“Of all the genes studied, significant differences in BMPRII expression were only found one month after initiation of supplementation in those patients who were not receiving natalizumab. A trend toward overexpression of BMPRII was observed in non–natalizumab-treated patients compared to natalizumab-treated patients, both at baseline and after six months. With respect to BMP5, we observed a decreasing trend in mean expression from baseline to six months in natalizumab-treated patients versus an increasing trend in non–natalizumab-treated patients. However, these changes were not statistically significant. The fact that there is an effective blockade of cell entry into the CNS depending on treatment with natalizumab and, therefore, apparently different CNS compartments, does not seem to be determinant in modifying TGF-β gene expression, except in BMPRII. Nonetheless, it cannot be excluded that the other mechanisms of action of DMT will affect gene expression of TGF-β in regards to VD.”

“This finding would be in line with other works, such as a study in which the authors measured expression of BMP genes in MS lesions and found increased expression of BMP2, BMP4, BMP5, BMP7, BMPRII, and pSMAD1/5/8 in astrocytes, microglia/macrophages, and neurons [22]. In another study, in this case in EAE, overexpression of Bmp4, Bmp6, and Bmp7 was observed in the lumbar spinal cord of mice with active EAE [40].”

  1. The conclusion is inadequately presented and does not effectively summarize the study's findings or their implications. Revise the conclusion to:
  • Succinctly recapitulate the main findings of the study, emphasizing the regulatory effect of vitamin D on specific genes in the TGF βpathway.

“VD indirectly regulates the in vivo expression of genes related to the TGF-β signaling pathway in RRMS patients. Increased expression of SMAD7 and ERK1 suggests that VD could affect both SMAD and non-SMAD signaling pathways, which are critical for the dual functions of TGF-β in the immune system and in MS”

  • Discuss the potential clinical implications of these findings for RRMS management and treatment strategies.

“Knowledge of the mechanisms of action of VD may yield biomarkers of response.”

  • Highlight the broader significance of the study in advancing our understanding of the complex interactions between vitamin D, immune regulation, and RRMS pathogenesis.

“Increased expression of SMAD7 and ERK1 suggests that VD could affect both SMAD and non-SMAD signaling pathways, which are critical for the dual functions of TGF-β in the immune system and in MS”

  • Clearly state the study's limitations and propose avenues for future research, such as exploring the molecular mechanisms underlying the observed gene expression changes.

“Our study could be limited by the impact of DMT mechanisms of action on gene expression and the evaluation of other subtypes of peripheral blood mononuclear cells. Further studies in larger cohorts are needed to elucidate the mechanisms by which VD exerts this regulation.”

  1. The manuscript requires extensive language editing to improve its academic style, coherence, and clarity. Pay attention to sentence structure, grammar, punctuation, and appropriate use of terminology throughout the manuscript.

Comments on the Quality of English Language

Although, the authors stated the manuscript has been revised by native speaker, the manuscript still contains merit of grammatical mistakes.

The manuscript has undergone a second round of review by a native-speaking medical writer with extensive experience in the field.

Round 3

Reviewer 2 Report

the authors have adequately addressed all concerns that have been raised by the reviewer

still need a moderate editing

Author Response

The manuscript has been thoroughly revised by a highly experienced medical writer, and no editing errors have been identified. I kindly request the reviewer to provide specific feedback on any inaccurate expressions or sentences that require further revision. Your detailed feedback will be immensely valuable for the enhancement of the manuscript.